# Smartphone-Based Prediction Model for Postoperative Cardiac Surgery Outcomes Using Preoperative Gait and Posture Measures

**DOI:** 10.3390/s21051704

**Published:** 2021-03-02

**Authors:** Rahul Soangra, Thurmon Lockhart

**Affiliations:** 1Crean College of Health and Behavioral Sciences, Chapman University, Orange, CA 92866, USA; soangra@chapman.edu; 2Fowler School of Engineering, Chapman University, Orange, CA 92866, USA; 3School of Biological and Health Systems Engineering, Arizona State University, Tempe, AZ 85287, USA

**Keywords:** frailty prediction models, regression models, postoperative outcomes, cardiac surgery, wearable sensors, smartphone apps

## Abstract

Gait speed assessment increases the predictive value of mortality and morbidity following older adults’ cardiac surgery. The purpose of this study was to improve clinical assessment and prediction of mortality and morbidity among older patients undergoing cardiac surgery through the identification of the relationships between preoperative gait and postural stability characteristics utilizing a noninvasive-wearable mobile phone device and postoperative cardiac surgical outcomes. This research was a prospective study of ambulatory patients aged over 70 years undergoing non-emergent cardiac surgery. Sixteen older adults with cardiovascular disease (Age 76.1 ± 3.6 years) scheduled for cardiac surgery within the next 24 h were recruited for this study. As per the Society of Thoracic Surgeons (STS) recommendation guidelines, eight of the cardiovascular disease (CVD) patients were classified as frail (prone to adverse outcomes with gait speed ≤0.833 m/s) and the remaining eight patients as non-frail (gait speed >0.833 m/s). Treating physicians and patients were blinded to gait and posture assessment results not to influence the decision to proceed with surgery or postoperative management. Follow-ups regarding patient outcomes were continued until patients were discharged or transferred from the hospital, at which time data regarding outcomes were extracted from the records. In the preoperative setting, patients performed the 5-m walk and stand still for 30 s in the clinic while wearing a mobile phone with a customized app “Lockhart Monitor” available at iOS App Store. Systematic evaluations of different gait and posture measures identified a subset of smartphone measures most sensitive to differences in two groups (frail versus non-frail) with adverse postoperative outcomes (morbidity/mortality). A regression model based on these smartphone measures tested positive on five CVD patients. Thus, clinical settings can readily utilize mobile technology, and the proposed regression model can predict adverse postoperative outcomes such as morbidity or mortality events.

## 1. Introduction

In the United States, one of the most commonly performed operations is cardiac surgery, and these surgeries have become routine in many centers [1]. It has also been demonstrated in several studies that although early postoperative complications and death are significant in the elderly undergoing cardiac surgery, excellent long-term outcomes can be achieved in selected populations [2,3,4]. The challenge to care providers has been determining what factors may predispose the older person to increased risk, realizing that no two are alike. Gerontology experts agree that frailty carries high vulnerability and a greater likelihood for poor health outcomes, including increased dependency, increased fall-risk, need for long-term care, disability, and mortality [5]. Frail individuals have difficulty maintaining physiologic balance when presented with stressors and have less ability to tolerate medical procedures or hospitalization. Although frailty is a significant predictor of disability and other adverse outcomes in older adults, it is also a risk factor for cardiac surgery patients. Furthermore, many cardiac clinicians do not know how to measure frailty, believe measurement to be unfeasible in the clinical setting, and have concerns that tools for measuring frailty have not been adequately compared for their usefulness in identifying frailty [6].

Recently, researchers have focused on finding specific performance measures that indicate frailty, particularly those suitable for the acute care arena. As a clinical marker for frailty, gait speed was determined to be the strongest predictor of mortality in individuals with coronary artery disease [6,7]. Afilalo et al. (2010) found that 5-m gait speed was associated with a 2- to 3-fold increase in the risk of significant mortality and morbidity in elderly patients undergoing cardiac surgery [8]. The Society of Thoracic Surgery’s (STS) Adult Cardiac Surgery Database recommended adding gait speed information to the database. The addition of gait speed will provide vital information to assist with cardiac surgery decision-making (STS, 2011).

This will also facilitate existing risk analysis by increasing the predictive value of mortality and morbidity following cardiac surgery for the elderly. This test is conducted with a stopwatch and a measured distance and provides a crude, single dimension of gait assessment. Additional measures are available for a more robust evaluation of gait and postural control parameters that may provide other data to be factored into the risk assessments. 

The purpose of this study was to improve clinical assessment and prediction of mortality and morbidity among older adults undergoing cardiac surgery through the identification of the relationships between gait and postural characteristics as measured by a smartphone device. This study utilized a proven non-invasive mobile sensor technology to accurately assess “frailty” using a self-contained gait and posture analysis system. This may allow a person’s gait and posture-related parameters to be measured in the clinic unobtrusively and enable the translation of these analyses to a monitoring and diagnostic tool that will allow tracking of gait and posture to quantify the frailty associated with a physiological degradation.

Our goals were to determine whether gait speed and postural stability (obtained via smartphone) correlate with major morbidity and mortality in a prospective sample of elderly patients undergoing cardiac surgery, and determine which measure is most predictive of postoperative morbidity and mortality. In this study, we developed a regression model for the prediction of adverse postoperative outcomes using smartphone derived measures of gait and postural stability. We tested the model on five patients who were not part of the model. To our knowledge, this is the first attempt to use mobile sensor technology to predict postoperative cardiac surgical outcomes.

## 2. Materials and Methods

The prospective study included 16 ambulatory patients 70 years of age or older undergoing non-emergent cardiac surgery. In a preoperative setting, patients were instructed to perform the 5-m walk and stand for 30 s in the clinic while wearing a smartphone device embedded with an inertial measurement unit. All physicians who were treating and patients were blinded to smartphone gait and posture assessment results to not influence the decision regarding surgery or postoperative management. Patient outcomes (defined by STS) were collected through the 30-day postoperative physician visit, at which time data regarding outcomes were extracted from the records. The anthropometric information for the sixteen cardiovascular disease (CVD) patients has been provided below in Figure 1. All patients provided written consent before participation, they were scheduled for cardiac surgery the next day. All patients were cognitively able to understand and follow instructions and could ambulate without any support. Patients were categorized as frail if they walked slower than 0.833 m/s, and otherwise categorized as non-frail [8]. The patient population for model creation consisted of five females and eleven males.

**Instrumentation:** A smartphone (with inbuilt inertial sensors) was affixed at the pelvic region using a smartphone holster and clip. Inertial sensor signals were sampled at 50Hz using the customized smartphone app “Lockhart Monitor” and used Matlab (MATLAB version 6.5.1, 2003, The MathWorks Inc., Natick, MA, USA) routines for data processing. The app was designed after consultations with human factors specialists at Virginia Tech and clinical requirements from registered nurse specialists in Carilion Roanoke Memorial Hospital (CRMH). The mobile app’s designed user interface consisted of a start button and stop button and contained recorded voice instructions with extended rest duration in-between each sequentially performed activity, as shown in Figure 2. 

**Experimental Procedure:** In this study, patients who were scheduled for cardiac surgery were screened by the registered PST nurse for inclusion. Only eligible patients were included in the study, and the PST nurse went over the written consent and explained the details of the experiment. If the patient consented to participate, a consenter (registered nurse specialist) answered all relevant questions about the study. We obtained written consent from participants according to Carilion Institutional Review Board (IRB).

Patients were instructed to wear a waist belt and smartphone was affixed around right ASIS. The walking area was marked at 0 m (start) and 5 m (end) and was well-lit. All patients followed voice instructions from the phone app and performed sit-to-stand and 5 m walk (Figure 2). Stopwatch was started and stopped as per first footfall when crossing 0 m and 5 m mark. The participants were instructed to repeat for three trials of sit-to-stand postural transition followed by postural stability and walking trials with appropriate rest in between each trial [9,10]. The smartphone app had two data collection modules: (1) 5 m walk; (2) STS and postural stability. The accelerometer signals were further processed using a zero-lag, low-pass butter-worth filter at cut off frequency of 6 Hz. This study’s independent variable is frailty (frail and non-frail as assessed by gait speed for the 5 m walk) utilizing stopwatch, and dependent variables are the adverse postoperative outcome (mortality and morbidity).

**Primary Outcome Measures** [8]: postoperative mortality and morbidity were the dependent variables. Morbidity among CVD patients was stroke, prolonged ventilation, deep sternal wound infection, renal failure, and reoperation [10]. Dependent variables are several traditional gait and posture measures, which have been reported in a multitude of studies. In this study, we used temporal and transitional aspects of gait (walking velocity) over several walking steps to assess/differentiate mobility decrements associated with frailty. The collection of these variables will allow useful comparisons to evaluate the effects of frailty on gait characteristics. We measured both walking speed and postural stability as a proxy for “frailty.” Walking speed was chosen because it is an independent predictor of “frailty” leading to morbidity and mortality [8]. Walking speed <0.65 m/s is considered as frail [11]. Gait speed [m/s] was computed using inertial sensors embedded inside the smartphone for a 5m long walk [9]. We calculated the resultant acceleration from all three directions of acceleration. The resultant acceleration signals were filtered using zero-lag 4th order low-pass Butterworth filter (cut-off frequency = 6 Hz). A half-second moving window variance was computed and the threshold was set using initial stand-still data as described in our previous studies [9,10]. Once the time to start and stop is detected, the average velocity can be computed over the 5 m walk.

Furthermore, to veer away from traditional linear tools that may mask the true structure of motor variability, a nonlinear dynamical analysis was applied to quantify movement variability/stability. Stability is undoubtedly a critical component of walking and balance and, is defined as the ability to maintain dynamic equilibrium despite the presence of small kinematic (movement) disturbances or control errors. Frailty has been associated with insufficient stability during quiet standing and has been assessed traditionally by the method of posturography (i.e., using a forceplate). In this study, we measured postural stability using the smartphone as a proxy for “frailty”. 

Linear variables like root mean square (RMS) are essential to measure accelerations’ fluctuation during the 5 m walk trial [12]. Three linear variability parameters were calculated using accelerations from three directions: root mean square of walking accelerations in the anterior-posterior direction (RMS_AP), the vertical direction (RMS_V), and medial-lateral direction (RMS_ML).

**Approximate Entropy (ApEn):** ApEn is an estimate of entropy and quantifies the system’s tendency to visit multiple states instead of remaining in preferred states [13]. This can be evaluated using acceleration (ACC) time series, where a moving window could compute the probability of short data short sequences of repetition within bounded tolerance throughout the ACC time series. In this study, we employed ApEn to quantify variability in ACC signals generated during a 5 m walk in CVD patients. ApEn has been reported to detect subtle changes in Center of Pressure (COP) variability that are not apparent sometimes in biomechanical linear measures of postural stability [14,15]. Pincus [16] reported the ApEn algorithm for complexity assessment in times series data. We applied acceleration time series (ACC) data for ApEn assessment. ApEn is defined as the logarithmic likelihood that the patterns of the given data near each other will remain nearby in a longer pattern. Given a sequence of total N numbers of ACC time series (x, y or z coordinate) like ACCx(1), ACCx (2), …, ACCx (N), similarly for ACCy(1), ACCy (2), …, ACCy (N) or ACCz(1), ACCz (2), …, ACCz (N). We used ACCx, ACCy or ACCz data set for ApEn in vertical, medial-lateral and anterior posterior directions. We divided the ACC timeseries of length N into short vectors of length m such as [*A*_m_ (1), *A*_m_ (2), …, *A*_m_ (N−m+1)], where the index i is ranging from 1 to N − m + 1. Where the distance between two vectors am (i) and *A*_m_ (j) is defined by |*A*_m_ (j) − *A*_m_ (i)|,
(1)$$C_{i}^{m}\left(d\right)=\frac{1}{N-m+1}\;\mathrm{such}\;\mathrm{that}\;\left|A_{m}\left(j\right)-A_{m}\left(i\right)\right|<d$$

The pattern length (m) was chosen as 2, the similarity coefficient (d) was set as 0.2% of the standard deviation of ACC data (collected from start to stop during 5 m walk at sampling frequency of 50 Hz) which has been shown two produce reasonable statistical validity of ApEn [10,17]. C_i_^m^(d) is the mean of the fraction of patterns of length m that resemble the pattern of same length that begins at index i. Finally, ApEn can be computed as
(2)$$\mathrm{ApEn}\left(N,m,d\right)={\left(N-m+1\right)}^{-1}\overset{N-\left(m-1\right)}{\underset{i=1}{\sum }}{\mathrm{lnC}}_{i}^{m}\left(d\right)-{\left(N-m\right)}^{-1}\overset{N-m}{\underset{i=1}{\sum }}{\mathrm{lnC}}_{i}^{m+1}\left(d\right)$$

The value of ApEn is unitless and is in range from 0 to 2 [13]. The pattern length (m) and tolerance level (r) are selected based on previous work on ApEn [15,18,19]. There will be few similar vectors of length m+1 than of length m, if the ACC data time series jumps around randomly. In contrast, if there are as many similar vectors of length m + 1 and m, then the ACC data time series has a repeating pattern to it. A smaller value of ApEn denotes a higher probability of regular repeating sequences. An ApEn value of zero, depicts that the time series is perfectly repeatable (such as a periodic sine wave or cosine wave), whereas the highest value of 2 is produced when the data is a random time series, for which repeating sequences only occur by chance (example Gaussian noise). All data processing for ApEn computation was performed using a customized MATLAB algorithm (Mathworks, Natick, MA, USA) for the AP and ML and vertical directions of ACC time series signals. 

**Statistical Analysis:** A multiple linear regressions (MLR) model was used to analyze relationship between dependent variables from gait and postural task with frailty. This method was chosen due to simplicity, small sample size of this special population, and its potential usage for quick frailty assessment in clinical environments before surgery. The variables were checked for normality, linearity, heteroscedasticity, and multicollinearity (variance inflation factor less than 5). A simple linear relationship was modeled with various dependent variables taken from 16 participants and the model was tested on 5 participants. 

## 3. Results

We found that the walking velocities computed using stopwatch time and smartphone time were correlated with Pearson correlation coefficient = 0.8154 and spearman’s rho = 0.8834 [10]. This study found eight participants as frail and eight as non-frail (walking velocity < 0.833 m/s for frail and walking velocity ≥ 0.833 m/s for non-frail). For ID04, ID14, and ID18 we found contrasting results for identifying frailty using smartphone and stopwatch. The surgical procedures carried out in 16 patients are tabulated in Table 1. The risk scores from STS are also provided in Table 1. For the smartphone-based velocity predictions of frailty, the specificity was 91.3% and a sensitivity of prediction as 79.2%. We found that the mean gait speed for frail was 0.67 m/s and that the non-frail group was 0.98 m/s. 

We have earlier reported significant differences among walking velocity, mean sway radius, sway area, sway path length, and mean sway velocity among frail patients compared to the non-frail counterparts [10]. These variability parameters have the potential to predict postoperative adverse outcomes among CVD patients. Utilizing important predictor variables new prediction models for postoperative cardiac surgical outcomes can be created. These prediction models can act as a clinical decision support system for assessing patients quickly in the perioperative period, and as such predictions for adverse postoperative outcomes can be made. 

This information allowed us to explore the association between postoperative outcomes and preoperative gait and posture measures. We expect that the acquisition of gait and postural characteristics will enable more accurate identification of frail older persons at higher risk of adverse postoperative conditions. These results provide vital information to assist with decision-making regarding cardiac surgery. 

We have earlier reported that postural sway parameters determined from smartphone [21], such as SD sway in anterior posterior, medial-lateral and resultant sway, along with linear variability measures such as RMS are significantly different in frail versus non-frail [10]. All sixteen patients were tracked after their surgeries for postoperative outcomes. We found that postoperative outcomes in CVD patients consisted of both morbidity and mortality. Two of the frail patients had a stroke (ID06 and ID21), three of the frail patients (ID08, ID11, and ID21) were kept for prolonged ventilation in the hospital, one frail patient (ID11) had a renal failure, one frail patient (ID21) was re-operated due to surgical complications, 3 frail patients (ID06, ID11, and ID22) and 1 non-frail (ID23) were sent to a skilled nursing facility, ID23 opted for skilled nursing facility due to lack of any caregiver. Only one of the frail patients (ID11) had a length of stay for more than 14 days, and one frail patient had mortality (ID21) [10]. 

Smartphone-derived variability measures such as RMS were significantly different in frail and non-frail patients. Non-frail patients produced substantially higher RMS-AP (*p* < 0.02), RMS-V (*p* < 0.01), and RMS-ML (*p* < 0.02) when compared to frail patients (Table 2).

Common postoperative outcomes observed in CVD patients were classified as morbidity and mortality. Morbidity consisted of: (i) stroke (either ischemic or hemorrhagic due to neurological deficit more than 72 h); (ii) renal failure (it is due to increase in serum creatinine more than 153 micromoles per liter or more than 2 mg/dL); (iii) prolonged ventilation for more than 24 h; (iv) deep sternal wound infection (DSWI); (v) reoperation for any surgical complication after 24 h of initial surgery; (vi) admissions to skilled nursing facility for rehabilitation; and (vii) prolonged length of stay in hospital more than 14 days postoperatively.

Linear measures of variability such as RMS in the vertical direction during a 5 m walk could classify frail versus non-frail with 100% sensitivity and 100% specificity [10]. Other variability measures (SD and RMS) derived during postural stability data collection from a smartphone could classify frail and non-frail with a specificity of 93.7% and sensitivity of 50%. Whereas only a cut-off point being 6.4547 mm in SD could classify frailty with a specificity of 93.7% 

Systematic evaluations of smartphone derived gait measures such as walking velocity (*p* = 0.009), root mean square accelerations in anterior-posterior (RMSAP) (*p* = 0.029), vertical direction (RMSV) (*p* = 0.013), and inter-step time (*p* = 0.034) and nonlinear postural measures of sway complexity in medio-lateral (ApEnMLCOP) and resultant direction (ApEnRCOP) were most sensitive to differences in groups Table 3. The postoperative adverse score prediction model was derived as shown in Table 3, using patient postoperative outcomes scored as per the criteria provided in Table 3. The model tested positive on five randomly chosen patients excluded from the model. Interactive dot-diagrams with high morbidity classification sensitivity were found with smartphone walking velocity cut-off ≤0.716 m/s, root mean square vertical acceleration cut-off ≤0.137 g, inter-step time cut-off >0.533 s, and approximate entropy in resultant sway direction with cut-off ≤1.139 (Figure 3). 

The model was tested on five CVD patients and their surgical procedures are presented in Table 4.

The model predicted fewer scores for patients ID02, ID05, and ID07 (less than 3) which were also not scored in STS risk rating. ID12 and ID25 rated higher by the model, and this corroborated with STS rating (Table 4). Both ID12 and ID25 had been admitted to a skilled nursing facility (high morbidity).

## 4. Discussion

In older CVD patients, there exists heterogeneity of health status (frail and non-frail) and this can lead to increased risk of postoperative complications [22,23], and thus surgical decision-making is challenging but critical for clinicians. For instance, if an older CVD patient is found with a high risk of adverse postoperative outcomes, then surgery can be delayed until the patient can rehabilitate to better health status (non-frail). Thus, preoperative risk assessment is essential. Unfortunately, there is a lack of tools for predicting operative risk. With the advancement of digital health technology for high fidelity movement data and fast computing, it is possible to deploy prediction models for clinical decision making. This is the first study developing a simple regression model from 16 CVD patients and testing the model on five randomly chosen CVD patients who underwent cardiac surgery and were excluded from the model. This study’s important contribution is to devise new simple methods based on mobile technology, which can impact clinical decision-making before surgery and reduce postoperative adverse outcomes among CVD patients. This study is novel since no previous research has developed a prediction model based on gait and posture movement characteristics; however, clinicians are overly dependent on “gait speed” as the only predictor for frailty and surgical outcomes.

Frailty is undoubtedly a critical marker of decreased physiologic reserves and resistance to stressors [24,25,26,27] and has been reported to predict operative risk in older surgical patients [23]. There is no standardized technique to measure physiologic reserve, but these may be critical to recovering from cardiovascular surgery [23]. The association of frailty with postoperative adverse outcomes is well known [28]. Our study utilizes knowledge from previous studies from Affilalo and coworkers, suggesting gait speed is an essential indicator of frailty [8,29,30,31] and adverse postoperative outcomes. As an advancement in knowledge, in our previous study, we reported that inertial sensor-based variables are indicators of frailty and postoperative outcomes in CVD patients [10]. Here in this study, we establish a relationship between smartphone-based variables and develop a simple regression model to predict postoperative outcomes. The simple model relies on gait and postural stability variables and is tested on five patients who were excluded from this model. We also posit that age may be a factor for operative complications [22,28,32,33,34,35], but one’s biological age could be a better representative of comorbid outcomes rather than chronological age [36,37,38,39]. 

In this study, we developed a simple regression model utilizing smartphones armed with sensors and linear/nonlinear variability analysis tools capable of predicting adverse postoperative outcomes in CVD patients. The essence of this study was that it was conducted in a clinical environment using a smartphone in CVD patients and led to the development of a postoperative outcome prediction model. 

The Society of Thoracic Surgeons (STS) promotes the use of quick tests, which can be practiced in clinics such as gait speed for the assessment of frailty among cardiovascular patients. Frailty among CVD patients is reported to be predictive of adverse health outcomes, including falls, institutionalization, hospitalization, and mortality [24,40,41]. Frail individuals are at high risk for falls, fractures, and hospitalizations, leading to death compared to non-frail counterparts [24]. Undoubtedly, STS guideline’s gait speed is an important measure, particularly among preoperative cardiac patients [8,42,43] and can utilize ubiquitous smartphones for its quick assessments [9,10]. Therefore, we categorized patients as frail and non-frail using the 5 m walk gait speed in this study. This study has established a relationship between posture and gait signatures of cardiac patients and their postoperative outcomes using a simple regression model. We have previously validated using inertial sensors in clinical environments [9,10] in end-stage renal disease patients [44,45]. An objective, accurate, and reliable way is required to assess frailty and postoperative outcomes without overly depending on a single gait measure such as “gait speed.” To achieve this goal, we devised using a smartphone with embedded inertial sensors that capture patients’ walking characteristics in clinical environments and their postural sway. Short bouts of gait such as five-meter gait speed certainly do not introduce fatigue in patients with cardiovascular impairments awaiting surgery [43]. But depending on the level of frailty, some patients who are expecting operations may not be healthy enough to walk on their own for 5 m. In such scenarios, it is worthwhile to examine the effects of postural control for balance and transitioning. We have previously reported that in the 5 m walking trials, the stopwatch time and smartphone time were highly correlated [10]. It was also found that smartphone-based postural and gait measures depicted adverse postoperative outcomes in CVD patients. In support of this hypothesis, we developed an adverse outcome prediction model (Table 3) with important gait (RMS-AP, RMS-V, and Step Time) and posture (ApEn-MLCOP, ApEn RCOP) variables from the smartphone. 

We found a significant increase in linear parameters of gait variability such as RMS-AP and RMS-V, along with step time intervals, helped differentiate frail versus non-frail patients. Coherently, it was also seen in nonlinear variability measures of postural stability like ApEn of mediolateral COP time series and ApEn of resultant COP time series were important in distinguishing frail versus non-frail patients. Frail patients had significantly lower complexity than the non-frail patients in COP sway. 

Nonlinear measures of postural signals are necessary and reveal subtle temporal properties of signals that are missed through traditional linear variability approaches [15,46,47]. Traditionally, higher COP excursions are linked with less postural stability, but since human balance maintenance is intrinsically complex and COP excursions do not holistically account for the system’s time-dependent evolution. We have utilized nonlinear measures such as complexity or approximate entropy to indicate deficient postural stability. Contrastingly, a healthy postural control system being more complex can readily adapt to unexpected perturbations to maintain balance. We found that frailty led to a loss of complexity of postural sway signals mainly in medio-lateral direction. 

Entropy-based estimations of postural sway signal irregularity and temporal structural organizational variability represent frail/non-frail participants’ adaptive capacity to maintain balance. Our findings coincide with previous investigations [13], which reported decreased complexity attributed to pathology and aging [48]. Frailty has probably reduced the degree of freedom of human postural control in the ML direction. Hence, the human postural control system cannot adjust to the demands inherent to frailty, thereby diminishing adaptability and stability of movement. CVD patients who are frail also have a low physiological reserve which could have probably led to decreased complexity due to impaired feedback control, impaired proprioception, or reduced muscle mass and strength, thereby leading to the reduced adaptive capacity during postural control [49]. Additionally, age-related decrements in sensory and neuromuscular control mechanisms could have exacerbated this problem [50]. 

From a clinical perspective, the postoperative outcomes and severity from high to low are in the order as death, prolonged ventilation, prolonged length of stay, discharge to a skilled nursing facility, stroke or renal failure, reoperation, deep sternal wound infection. The simple regression model was tested on five patients (ID02, ID05, ID07, ID12, and ID25) and their respective model scores were 2.61, 1.64, 1.34, 8.63, and 4.34. ID12 (model score 8.63) and ID25 (model score 4.34) who had the highest model score among all five patients were admitted to a skilled nursing facility. The remaining three patients ID02, ID05, and ID07 who had model scores of 2.61, 1.64, and 1.34 respectively, also were not scored for STS risk score. 

Unwittingly, gait measures and associated linear variability as measured by root mean square (RMS) of acceleration signals may be optimal from energy expenditure [51], temporal [52], or spatial variability [53], or the perspective of attentional demands [54]. Walking stability is crucial since almost 70% of total falls occur during locomotion [55,56]. Moe-Nilssen evaluated walking stability using inertial sensors at the trunk level [57,58] and reported higher average accelerations or fluctuations in people with balance impairments [59]. Gait measures are crucial, since previous authors have reported gait speed as an independent sign of adverse postoperative outcomes. During locomotion, humans have to respond to multiple unexpected perturbations generated during walking. Thus, both postural and gait measures are critical in the adverse postoperative outcome predictive model, preferably an element of a healthy pace and postural control system, which can adapt to unexpected perturbations in an attempt to maintain healthy recovery after surgery. We tested the regression model (Table 3) on five CVD participants and found that the difficulty level of surgical procedures (Table 4) was in coherence with model output scores (Table 5) utilizing a smartphone app. High model output scores were correlated with increased morbidity as both ID12 and ID25 were admitted to a skilled nursing facility.

In this study, a 5 m walk was measured using smartphone accelerometers situated at the pelvis, and these accelerations generated depicted stability of the participants countering internal (related to CVD) and external (environment-related) perturbations. We walk at a preferred speed, which is a combination of step length and step frequency and is an essential factor in controlling dynamic balance [60]. This preferable speed was selected to optimize the stability of the walking pattern. We found that fluctuations in accelerations as measured by RMS were significantly different in frail and non-frail patients. But RMS acceleration may be correlated with walking speed, which is different among frail and non-frail counterparts. Frail patients preferred a slower walking pace than the non-frail patient to minimize acceleration fluctuations or RMS values. Thus, it was found that frail patients produced significantly higher accelerations and were unable to provide smooth and rhythmic movements accelerations at trunk level while walking at slower velocities. Our model suggested that smartphone walking speed along with postural measures from a smartphone such as ApEn of COP in ML and resultant (R) and gait measures of the smartphone such as RMS AP and RMS Vertical and step duration are predictive of frailty with high accuracy (Table 3). These frailty indicators and simple regression models could be used as prescreening tools for cardiac surgical procedures and help clinicians identify frail patients who may need intensive rehab or preplan their stay in hospital with specialized nursing care before their return to home. 

Thus, simple classification models empowered with posture and gait signatures from inertial sensors embedded inside smartphone have the potential to predict frailty and adverse postoperative outcomes in CVD patients, although a great deal of work is needed in future to make such research tools easy to use for clinicians. To meet the clinical challenges of decision making along with patient safety and point of care, new technologies are needed which can be used in clinical environments without hindering medical routine for patients and hospital staff. This study has attempted to provide a prediction model of postoperative cardiac surgical outcome using smartphone-based preoperative gait and posture measures. 

## 5. Limitations

The size of the population limits the prediction model. The model was built using patient data from sixteen patients and was tested on five other patients. The study was conducted in a clinical environment, and patients were aware that they were participating in a frailty assessment protocol using a smartphone. The performance of patients could have been affected by being conscious of the environment. Some patients had a high level of anxiety to some extent for their surgery allotted for the next day, and non-laboratory setting limited the scope of this data. Another limitation of this work is only a linear model was developed for frailty classification, a neural network with a single neuron could have resulted in a nonlinear regression prediction. The authors understand that with more data robust models based on neural network can be trained with high classification accuracy. In future, we plan to collect more data using smartphones and store in cloud for developing machine learning-based frailty classification models. However, initial data analyses may provide insights into how the study can be conducted in a larger population. The app designed for data collection and automatic gait speed estimation by smartphone required strict following of protocol. If any other movement artifact is found after or before the walking task, the movement detection time may be influenced, thus affecting gait speed estimation from the smartphone 

## 6. Conclusions

The prediction of frailty and postoperative outcomes in CVD patients is essential, and mobile technology-based predictive analysis can help in clinical evaluation and early decision making for cardiac patients. Earlier detection of patients at higher risk of major health-related events such as morbidity or mortality is crucial, since they impact patients’ stay in hospital and rehabilitation therapy. This study demonstrated that simple models based on gait and posture linear and nonlinear variability measures from a smartphone could reliably distinguish patients with a high risk of adverse postoperative outcomes. By empowering clinicians with smartphone-based clinically useful gait and posture measures which are simple, quick, easy to perform in clinics, it is hoped that the smartphone can help in early decision making. The findings suggest that various variability parameters in walking and stand-still posture can be easily implemented in clinics with high acceptability and a low risk. A simple predictive model has been developed in this study for predicting postoperative morbidity and in future, a study with a larger sample size can be conducted to improve the accuracy of these models.

## Figures and Tables

**Figure 1 sensors-21-01704-f001:**
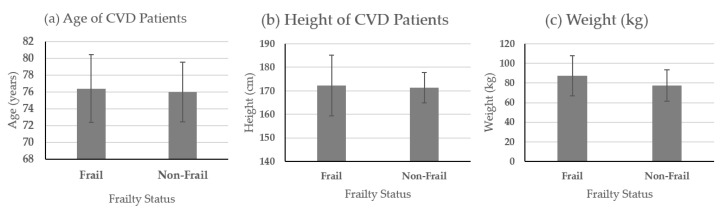
Patients’ anthropometric data. (**a**) Age in years, (**b**) patient height in centimeters, (**c**) patient weight in kilograms.

**Figure 2 sensors-21-01704-f002:**
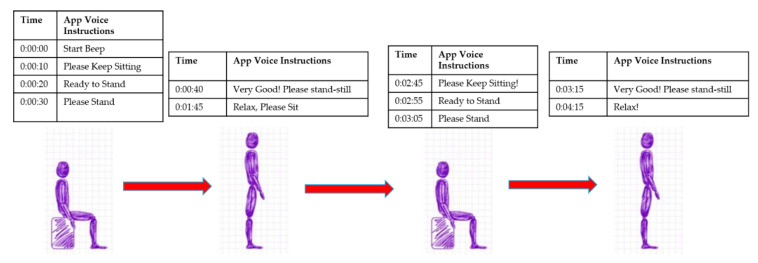
Smartphone App voice commands during data collection.

**Figure 3 sensors-21-01704-f003:**
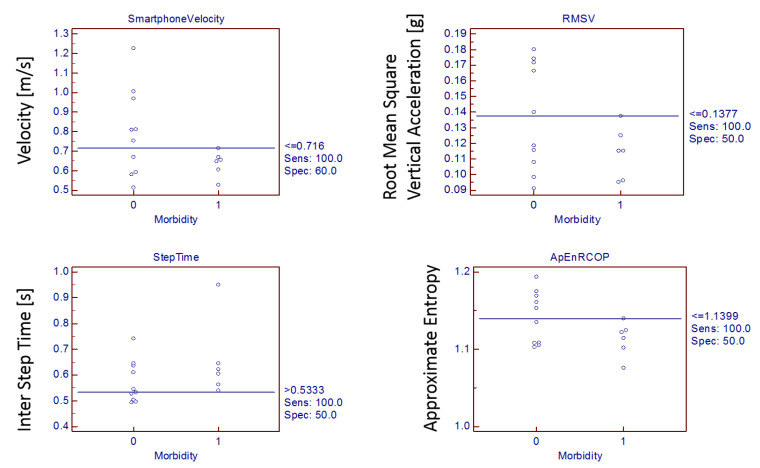
Interactive dot-diagram displaying highest sensitivity in morbidity classification (0-nonmorbid, 1-morbid) with smartphone walking velocity, root mean square vertical acceleration, inter-step time, and approximate entropy in resultant sway direction.

**Table 1 sensors-21-01704-t001:** Surgical procedures of all patients and their Society of Thoracic Surgeons (STS) risk scores.

ID	Surgical Procedure	STS Risk Score
**ID04**	CABG × 4	0.012
**ID06**	AVR, MV repair, CABG × 3, Maze	NS
**ID08**	AVR, CABG × 1	0.035
**ID09**	AVR, CABG × 2	0.013
**ID10**	AVR, CABG × 2	0.027
**ID11**	CABG × 2, Extensive Maze	0.021
**ID13**	AVR, CABG × 3	0.013
**ID14**	AVR	0.023
**ID17**	AVR	0.019
**ID18**	AVR, CABG × 2	0.042
**ID19**	AVR (re-do sternotomy)	0.044
**ID20**	AVR, Root Replacement, coronary reconstruction	NS
**ID21**	MVR, CABG × 1 (re-do sternotomy)	0.164
**ID22**	MV Repair	0.020
**ID23**	AVR, CABG × 1, limited concomitant Maze	NS
**ID24**	AVR, CABG × 3	0.014

CABG (Coronary Artery Bypass Grafting); AVR (Aortic Valve Replacement); MV Repair (Mitral Valve Repair); Maze Surgical Procedure (maze surgery cures Atrial Fibrillation by creating a “maze” of new electrical pathways to let electrical impulses travel easily through heart); Redo Sternotomy (A sternotomy done after a previous sternotomy, usually after significant scar tissue has formed); Root Replacement Coronary reconstruction (Specialized operation that repairs the portion of the aorta closest to the heart, while preserving the patient’s own aortic valve); STS score as per Society of Thoracic Surgeons National Database [20]. NS “not scored’ if the surgical complication did not fit in STS clinical practice guideline.

**Table 2 sensors-21-01704-t002:** Root mean square (RMS–AP, ML, V) from 5 m walk utilizing smartphone signals. SD and CV are standard deviation and coefficient of variation. Here * represents significant difference between frail and non-frail patients with *p* < 0.05.

Variables	Health Status					
	Frail			Non-Frail		
	Mean	SD	CV	Mean	SD	CV
RMS_AP *	0.12	0.03	24.50	0.15	0.02	15.23
RMS_V *	0.11	0.01	11.82	0.17	0.02	10.40
RMS_ML *	0.11	0.02	19.63	0.15	0.03	22.48

**Table 3 sensors-21-01704-t003:** Gait and posture measures of patients with morbid and non-morbid outcomes; and adverse outcome score prediction model; and Scoring criteria with weightage. Here * represents significant difference between frail and non-frail patients with *p* < 0.05.

(a)	No Morbidity		Morbidity		
	Mean	SD	Mean	SD	*p*-Value
**Stopwatch Velocity * [m/s]**	0.865	0.217	0.701	0.097	0.0078
**SmartphoneVelocity * [m/s]**	0.793	0.222	0.637	0.064	0.009
**RMSAP * [g]**	0.128	0.026	0.113	0.033	0.029
**RMSV * [g]**	0.136	0.034	0.114	0.016	0.013
**ApEnMLCOP**	1.077	0.068	1.101	0.018	0.335
**ApEnRCOP**	1.141	0.033	1.113	0.021	0.088
**StepTime * [s]**	0.573	0.081	0.654	0.150	0.034
**(b) Adverse Outcome Score**	Score = 44.13 − 26.96 × RMSAP − 28.29 × RMSV + 13.33 × ApEnMLCOP − 46.85 × ApEnRCOP + 5.35 × StepTime				
**Prediction Model**					
**(c) Scoring Criteria** **(Cp = 2.77, R-square = 0.52)**	Death = 7; Prolonged Ventilation = 6; Prolonged length of stay = 5; Discharge to Skilled Nursing Facility = 4; Stroke/Renal Failure = 3; Reoperation = 2; Deep Sternal Wound Infection = 1				

**Table 4 sensors-21-01704-t004:** Surgical procedures of all patients and their STS risk scores.

ID	Surgical Procedure	STS Risk Score
**ID02**	AVR, Root Replacement, coronary reconstruction	NS
**ID05**	AVR, Root Replacement, reimplantation of coronaries	NS
**ID07**	AVR, MV repair, CABG × 3, Extensive Maze	NS
**ID12**	MVR	0.062
**ID25**	AVR	0.018

**Table 5 sensors-21-01704-t005:** Regression model output scores for five cardiovascular disease (CVD) patients.

ID	RMS_AP	RMS_V	APEN_ML_COP	APEN_R_COP	STEP TIME	MODEL SCORE
**ID02**	0.093	0.118	0.991	1.105	0.550	2.613
**ID05**	0.105	0.107	0.126	1.174	0.650	0.648
**D07**	0.121	0.111	0.087	1.148	0.550	1.346
**ID12**	0.039	0.039	1.118	0.141	0.980	8.633
**D25**	0.111	0.137	1.097	1.076	0.540	4.340

## Data Availability

Not applicable.

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
