# Peer review of "Smartphone-Based Prediction Model for Postoperative Cardiac Surgery Outcomes Using Preoperative Gait and Posture Measures"

_sensors, 2021, doi:10.3390/s21051704_

Round 1
Reviewer 1 Report
In the submitted manuscript, the authors developed health-related metrics utilizing data provided by a smartphone (with embedded inertial sensors) attached to the participants’ waist to predict the mortality and morbidity among older cardiovascular disease (CVD) patients following major surgery. The metrics included walking velocity, step time, root-mean-square (RMS) acceleration (AP: anterior-posterior, ML: medial-lateral, V: vertical), and approximate entropy for all three acceleration directions. Metrics with significant differences between patients with morbid and non-morbid outcomes were included as predictor variables in a linear regression model predicting STS adverse outcome score. This model was trained on 16 participants’ data and tested on 5 others’ data.
There are a few major revisions that should be addressed prior to publication. First, there should be a subsection in the Materials and Methods section that explicitly describes the statistical analyses reported in the Results section. For example, it was not clear until lines 274-275 that the predictive model was trained on 16 participants’ data and validated on 5 randomly chosen participants’ data. Related to this point, can the authors comment on why they chose not to conduct a k-fold cross validation? Meaning split the population into k groups, train the model using data from k-1 groups, validate the model on the remaining group, and repeat for all combinations. Next, it would be helpful to see a graphic illustrating the number of patients who were categorized as frail or non-frail who had morbidities or not (basically, lines 230-236 in graphic form). This would highlight why walking speed alone is not sufficient to predict postoperative adverse outcomes, hence the need to include other predictor variables.
A few minor revisions are also listed below:
- On lines 200-202, the authors report both Pearson and Spearman correlation coefficients. Does this imply the relationship between the stopwatch time and smartphone time is monotonic but nonlinear? If so, can the authors comment on why the relationship between the times is noninear?
- On lines 203-205, can the authors identify whether ID04, ID14, and ID18 were characterized as frail or non-frail given the discrepancy between the stopwatch and smartphone? What was the deciding factor?
- In Table 1, can the authors confirm (and specify in the manuscript) that ‘NS’ stands for not scored? If so, can the authors provide a reason for why the patients were not scored in the table description or the body of text when the table is introduced? Same question for lines 276-277.
- In Table 2, can the authors confirm (and specify in the manuscript) if ‘CV’ stands for coefficient of variance?
- Could the authors comment on the difference between the cut-off walking velocity reported on line 265 (0.716 m/s2) and the classification threshold previously described/used (0.833 m/s2)?
- In figure 3, why display the cut-off with the highest sensitivity instead of the cut-off that maximizes the sensitivity and specificity?
- Non-frail patients produced higher accelerations during walking than frail patients, but frail patients exhibit less movement complexity.
- For the entirety of manuscript, the authors are encouraged to evaluate how many significant figures are necessary to report their quantitative data.
Overall, this manuscript was relatively well-written and communicated. It is the opinion of this reviewer that the manuscript be accepted after major revisions.
Author Response
Reviewer 1:
Thank you for giving us the opportunity to submit a revised draft of the manuscript “Smartphone-based Prediction Model for Postoperative Cardiac Surgery Outcomes using Preoperative gait and Posture Measures”. We appreciate the time and effort that you and the reviewers dedicated to providing feedback on our manuscript and are grateful for the insightful comments on and valuable improvements to our paper. We have incorporated most of the suggestions made by the reviewers. Those changes are highlighted within the manuscript. Please find below, in blue, for a point-by-point response to the reviewers comments and concerns.
Comments and Suggestions for Authors
Comment: In the submitted manuscript, the authors developed health-related metrics utilizing data provided by a smartphone (with embedded inertial sensors) attached to the participants’ waist to predict the mortality and morbidity among older cardiovascular disease (CVD) patients following major surgery. The metrics included walking velocity, step time, root-mean-square (RMS) acceleration (AP: anterior-posterior, ML: medial-lateral, V: vertical), and approximate entropy for all three acceleration directions. Metrics with significant differences between patients with morbid and non-morbid outcomes were included as predictor variables in a linear regression model predicting STS adverse outcome score. This model was trained on 16 participants’ data and tested on 5 others’ data.
Response: Thank you for your comment. We agree and have included this section now in the manuscript.
Comment: There are a few revisions that should be addressed prior to publication. First, there should be a subsection in the Materials and Methods section that explicitly describes the statistical analyses reported in the Results section. For example, it was not clear until lines 274-275 that the predictive model was trained on 16 participants’ data and validated on 5 randomly chosen participants’ data. Related to this point, can the authors comment on why they chose not to conduct a k-fold cross validation? Meaning split the population into k groups, train the model using data from k-1 groups, validate the model on the remaining group, and repeat for all combinations. Next, it would be helpful to see a graphic illustrating the number of patients who were categorized as frail or non-frail who had morbidities or not (basically, lines 230-236 in graphic form). This would highlight why walking speed alone is not sufficient to predict postoperative adverse outcomes, hence the need to include other predictor variables.
Response: Thank you for this comment. There are several advantages of k-fold cross-validation such as i) reduces overfitting and ii) tuning of hyperparameters during machine learning. Since our model was not built on machine learning algorithm and was instead a simple MLR with one predictive outcome, we are limited in applying k-fold cross validation. In addition, there is bias-variance trade-off associate with the choice of k. Typically, given the considerations, one performs k-fold cross-validation using k=5. We are limited with small sample size of 5 subjects for testing. We acknowledge concerns of reviewer that we have small sample size and may be limited with overfitted model for training data (n=16 subjects) and small testing size (n=5). We are grateful to reviewer for these suggestions. In future, we plan to implement on new data set collected using app and stored in cloud. A large number of subjects would be helpful to train model and testing for overfitting through cross-validation. We plan to utilize recurrent neural network (Long-short term memory) for gait and postural data from accelerometers during walking. We believe both gait and postural data will make this model more robust in classification for frailty or adverse post-operative outcomes.
Acknowledging important concerns from reviewer, we also would like to highlight here that this special clinical population (CVD patients who volunteered for participation before surgery), group of cardiac specialist, and engineering researchers was rare combination and the scientific information generated has high clinical value in saving lives and improving quality of life (QOL) of CVD patients. Authors want to highlight the high impact of this research in frailty intervention the importance of dissemination.
We have acknowledged this in the limitation section “Another limitation of this work is only a linear model was developed for frailty classification, a neural network with a single neuron could have resulted in a nonlinear regression prediction. Authors understand that with more data robust models based on neural network can be trained with high classification accuracy. In future, we plan to collect more data using smartphones and store in cloud for developing machine learning-based frailty classification models.”
Comment: A few minor revisions are also listed below:
On lines 200-202, the authors report both Pearson and Spearman correlation coefficients. Does this imply the relationship between the stopwatch time and smartphone time is monotonic but nonlinear? If so, can the authors comment on why the relationship between the times is noninear?
Response: Thank you for this comment. Yes, the relationship was monotonic and non-linear. Authors think the error may have induced non-linear behavior. However the errors were small therefore it was monotonic. Authors think this can be due to two reasons:
1) Delay by RN in using stopwatch: The Registered Nurse (RN) involved in this study was highly trained for evaluating 5-m gait speed. The RN noted the first and last footfalls when crossing the 5-m marks. Since time interval is being measured, the fixed neuromuscular delay from visual signal to the pressing of stopwatch is not important as long as it remains relatively constant. We do not think practice could have improved performance, since she was highly trained, but fatigue and caution for patient while walking could have impacted her response to some extent.
2) Delay by smartphone: Gait initiation involves a complex sequence of anticipatory postural adjustments (APAs) during transition from steady state standing before mark and walking. This will lead to early registering of gait initiation. Similarly, APAs may affect termination of gait with some delay in registering termination.
Comment: On lines 203-205, can the authors identify whether ID04, ID14, and ID18 were characterized as frail or non-frail given the discrepancy between the stopwatch and smartphone? What was the deciding factor?
Response: Thank you for this comment. Earlier studies have considered gait speed as a biomarker of frailty[1, 2] and have used stopwatch. But gait speed categorized subjects ID04, ID14 and ID18 as non-frail, however, these patients had adverse post-operative outcomes CABG (4 grafts in total) (for ID04), Aortic Valve Replacement (AVR) (for ID14) and AVR, CABG (2 grafts) (for ID18). They also had high society of thoracic surgeons (STS) scores i.e. 0.012, 0.023, and 0.042 respectively for high risk of mortality and morbidity. The post-operative outcomes and STS scores were the deciding factor.
Comment: In Table 1, can the authors confirm (and specify in the manuscript) that ‘NS’ stands for not scored? If so, can the authors provide a reason for why the patients were not scored in the table description or the body of text when the table is introduced? Same question for lines 276-277.
Response: Thank you for this comment. STS scores are evaluated before surgery to decide whether a patient should undergo surgical or transcatheter aortic valve replacement (TAVR). STS scores cannot be computed for complex procedures for which there is no STS clinical practice guideline. ID06, ID20 andID23 do not fit in available STS guideline thus could not be scored. We have included the reason for NS ‘Not Scored’ STS scores. Some online tools such as http://riskcalc.sts.org/stswebriskcalc/calculate and STS/ACC TAVR Risk Calculator App are available nowadays for STS score assessment.
Comment: In Table 2, can the authors confirm (and specify in the manuscript) if ‘CV’ stands for coefficient of variance?
Response: Thank you we have indicated it now in the manuscript.
Comment: Could the authors comment on the difference between the cut-off walking velocity reported on line 265 (0.716 m/s2) and the classification threshold previously described/used (0.833 m/s2)?
Response: Thank you for this comment. 0.833 m/s is the smartphone velocity as per the criteria of frailty proposed Afilalo and coworkers ( crossing 5-meters in 6 seconds)[2, 3]. Whereas we found in our dataset a cutoff smartphone velocity ≤0.716 could classify frailty with sensitivity of 100% and specificity of 60%.
Comment: In figure 3, why display the cut-off with the highest sensitivity instead of the cut-off that maximizes the sensitivity and specificity?
Response: Thank you! It is important to stop patients who are frail to go for surgery. This will save lives since frail individuals can be rehabilitated until they are no more frail and could be operated thereafter. Thus, sensitivity is of high importance and not specificity.
Comment: Non-frail patients produced higher accelerations during walking than frail patients, but frail patients exhibit less movement complexity.
Response: Yes, this is correct. Complexity decreased due to frailty making the patients less adaptable (more rigid) and more prone to fall. Non-frail patients had better health status thus could generate high accelerations during STS and walking.
Comment: For the entirety of manuscript, the authors are encouraged to evaluate how many significant figures are necessary to report their quantitative data.
Response: Thank you we have changed to 3 significant digits now.
Comment: Overall, this manuscript was relatively well-written and communicated. It is the opinion of this reviewer that the manuscript be accepted after major revisions.
Response: We appreciate your comments, these suggestions have improved the quality of this paper.
- Chen, M.A., Frailty and cardiovascular disease: potential role of gait speed in surgical risk stratification in older adults. J Geriatr Cardiol, 2015. 12(1): p. 44-56.
- Afilalo, J., et al., Gait Speed and 1‐Year Mortality Following Cardiac Surgery: A Landmark Analysis From the Society of Thoracic Surgeons Adult Cardiac Surgery Database. Journal of the American Heart Association, 2018. 7(23).
- Afilalo, J., et al., Gait Speed and Operative Mortality in Older Adults Following Cardiac Surgery. JAMA Cardiology, 2016. 1(3): p. 314.
Reviewer 2 Report
In this manuscript, authors study the relationships between preoperative gait and postural stability characteristics and postoperative cardiac surgical outcomes. The main work of the manuscript is to present a linear model (Table 3(b)) which is identified through simple linear regression. The problem is undoubtedly important but the contribution of the manuscript is not sufficient.
- Comparing with authors’ previous work, in particular [10], the majority of the results in this manuscript is very similar to [10] except the linear model.
- It is highly likely that the complex relationship between the frailty and gait and postural stability characteristics is nonlinear. There are many new techniques which can be used for classification and regression such as nonlinear regression, support vector machine and many clustering methods. The technical contribution of the manuscript is very limited.
- As the authors indicate, the size of the population is limited (16 subjects for model development and 5 for test). Actually the size for effective test is 2 (ID12 and ID25) is too small to give meaningful conclusion.
Author Response
Reviewer 2
Thank you for giving us the opportunity to submit a revised draft of the manuscript “Smartphone-based Prediction Model for Postoperative Cardiac Surgery Outcomes using Preoperative gait and Posture Measures”. We appreciate the time and effort that you and the reviewers dedicated to providing feedback on our manuscript and are grateful for the insightful comments on and valuable improvements to our paper.
Comment: In this manuscript, authors study the relationships between preoperative gait and postural stability characteristics and postoperative cardiac surgical outcomes. The main work of the manuscript is to present a linear model (Table 3(b)) which is identified through simple linear regression. Comparing with authors’ previous work, in particular [10], the majority of the results in this manuscript is very similar to [10] except the linear model.
Response: Thank you for this comment. We acknowledge that we are limited with small sample size of 5 subjects for testing. We also acknowledge concerns of reviewer that we have small sample size and are limited with overfitted model for training data (n=16 subjects) and small testing size (n=5). However, we would like to highlight here that this special clinical population (CVD patients who volunteered for participation before surgery), group of cardiac specialist, and engineering researchers was rare combination and the scientific information generated has high clinical value in saving lives and improving quality of life (QOL) of CVD patients. Authors want to highlight the high impact of this research in frailty intervention and the importance of dissemination.
We have acknowledged this in the limitation section “Another limitation of this work is only a linear model was developed for frailty classification, a neural network with a single neuron could have resulted in a nonlinear regression prediction. Authors understand that with more data robust models based on neural network can be trained with high classification accuracy. In future, we plan to collect more data using smartphones and store in cloud for developing machine learning-based frailty classification models.”
Comment: It is highly likely that the complex relationship between the frailty and gait and postural stability characteristics is nonlinear. There are many new techniques which can be used for classification and regression such as nonlinear regression, support vector machine and many clustering methods.
Response: Thank you for this comment. Acknowledging the limitations of this study, we think the importance of this work lies in its simplicity (simple model) which can be quickly used by clinicians in clinical environment using a smartphone app. Authors are working on building Apps which could be deployed to extract complexity and variability (RMS) in AP, ML and vertical directions. In future, authors plan to collect more data and develop supervised ML models.
Comment: As the authors indicate, the size of the population is limited (16 subjects for model development and 5 for test). Actually the size for effective test is 2 (ID12 and ID25) is too small to give meaningful conclusion.
Response: Thank you for this comment. STS scores are evaluated before surgery to decide whether a patient should undergo surgical or transcatheter aortic valve replacement (TAVR). STS scores cannot be computed if STS clinical practice guideline is not present for certain cases. Since the patients have different combinations of cardiovascular disorders, the scoring for ID02, ID05 and ID07 were not available as per the STS guidelines. But this does not limit our conclusions of post-operative morbidity. Since ID12 and ID25 were admitted to skilled nursing facility, they certainly had severe morbidity compared to ID02, ID05 and ID07.
Round 2
Reviewer 1 Report
The authors addressed all of my comments. It is therefore the opinion of this reviewer that the manuscript be accepted in its present form.
Reviewer 2 Report
I agree with authors' argument that this research is important in terms of frailty intervention and offering clinicians a simple model with a smartphone app with high sensitivity. The limitation of model robustness and small sample size needs to be improved in future study.